# The Preparation of a Superhydrophobic Fluorine Rubber Surface

**Xinyang He ***, **Shuaichang Ren and Ruiting Tong ***

School of Mechanical Engineering, Northwestern Polytechnical University, Xi'an 710072, China
* Correspondence: lotucell@163.com (X.H.); tongruiting@nwpu.edu.cn (R.T.)

**Abstract:** Superhydrophobic materials have a good application prospect in self-cleaning, anti-fouling, anti-corrosion, and anti-freezing. However, creating large areas of simple, efficient, and environmentally friendly superhydrophobic surfaces remains a huge challenge. In this paper, a simple, environmentally friendly surface superhydrophobic preparation method is used based on 107 silicone rubber adhesive. A superhydrophobic coating with a micro/nano structure was constructed on the surface of fluorine rubber. The particle size and groups of HB-192V powder (mainly SiO$_2$) were observed by Fourier-transform infrared spectroscopy and a scanning electron microscope (SEM). The structures of two kinds of rubber surfaces were observed by SEM, and the superhydrophobic surface materials were qualitatively analyzed by X-ray diffraction (XRD). The hydrophobic properties of the superhydrophobic surface and the bouncing properties of droplets were analyzed by a contact angle measuring instrument and a high-speed camera. The results show that the preparation of superhydrophobic fluorine rubber on the surface of the water meter contact angle could reach an average of 154.1°. The superhydrophobic surface prepared by this method has a better hydrophobic and droplet bounce property.

**Keywords:** superhydrophobic; fluorine rubber; HB-192V; bounce performance



## 1. Introduction

In recent years, superhydrophobic materials are a novel and popular object of research. Regarding the concept of superhydrophobicity, the contact angle is an important indicator to reflect the wettability of the material surface. A surface with a contact angle greater than 150° and a rolling angle less than 10° is called a superhydrophobic surface [1]. Superhydrophobicity exists widely in nature. In 1997, German botanist Barthlott [2] discovered self-cleaning and superhydrophobicity in lotus leaves that "emerge from silt without staining". Subsequently, more and more researchers are engaged in the study of superhydrophobicity. Superhydrophobicity has also been found in some animals, including water strider feet [3], mosquito eyes [4], fish scales [5], and butterfly wings [6].

To prepare superhydrophobic surfaces, two conditions should be met at the same time. First, the surface of the material should have a rough micro-nano structure, and second, the surface of the material should be modified with low-surface-energy substances [7]. At present, the relatively mature methods to prepare superhydrophobic surfaces mainly include the template method [8,9], the layer-by-layer self-assembly method [10,11], the deposition method [12], the electrostatic spinning method [13], the etching method [14], the phase separation method [15], and so on. A superhydrophobic surface has excellent hydrophobicity, and its special wettability makes superhydrophobic surfaces widely used in self-cleaning [16], anti-icing [17], anti-corrosion [18], oil-water separation [19], and other fields. At present, more and more scholars are conducting research in the field of superhydrophobicity and are trying to prepare superhydrophobic surfaces in novel ways. Zhang et al. [20] obtained a rough, three-dimensional surface structure by electrodepositing copper on nickel foam. In addition, the self-assembly of 1-octanethiol on a microstructure can

reduce the surface energy and protect the copper coating. Three-dimensional SHM shows a high degree of hydrophobicity. Tang et al. [21] prepared a solid micro-nano structure superhydrophobic surface for long-term dripping condensation through a microcapillary covered with nano grass, and expounded the application integration of this promising functional surface. Wu et al. [22] prepared a multifunctional superhydrophobic coating by compounding $Fe_3O_4$ nanoparticles with fluorinated epoxy resin by the method of reverse osmosis and found that the layer prepared by this method had outstanding hydrophobicity and icing-delay performance. Wang et al. [23], based on the structure of the surface of the lotus leaf, constructed a similar micro/nano rough structure on PDMS, made ZnO nanohairs grow on a polydimethylsiloxane (PDMS) surface by enhanced hydrothermal technology, and prepared a superhydrophobic surface by fluorosilane modification.

Fluorine rubber refers to a synthetic polymer elastomer containing fluorine atoms on the carbon atoms of the main chain or side chain. It has good high-temperature resistance, low-temperature resistance, corrosion resistance, and aging resistance, and good mechanical properties [24–26]. On the basis of its good comprehensive performance, fluorine rubber has been widely used in various fields [27] and is an indispensable basic material in aerospace, the automobile industry, national defense, military industries, the petrochemical industry, and other industries. In daily life, fluorine rubber is widely used in the automobile industry, such as a diaphragm and gasket on certain mechanical mechanisms, as an oil seal on an engine, and as the inner rubber layer of a fuel-hose composite structure, and so on. There are also many rubber products in oil drilling and production engineering [28], such as centralizers, rubber packers, and cementing rubber plugs, etc. Due to the particularity of their working environments, such industries have high requirements for rubber materials. Fluorine rubber is a good choice of rubber materials in oil drilling and production engineering because of its excellent oil and corrosion resistance. In terms of hydrophobic research of fluorine rubber, He et al. [29] prepared hydrophobic surfaces with micro and nano structures on the surface of fluorine rubber by using the template of a 900-mesh sieve using the method of surface hot pressing. Zhou et al. [30] reported the preparation and characterization of a fluororubber-coated superhydrophobic polyethylene tetramethyl ester fabric and polyurethane sponge. Its water contact angle is $\geq 153°$, and it has good wear resistance. So far, the research on superhydrophobic rubber is mainly focused on silicone rubber [31–35] and there is little research on superhydrophobic fluorine rubber, both at home and abroad. Moreover, the existing preparation methods of superhydrophobic surfaces are complicated in operation and environmentally unfriendly.

In this paper, a simple and convenient method of powder spraying was used to construct a layer of a silica superhydrophobic surface on the surface of fluorine rubber with 107 liquid silicone rubber as an adhesive. The constructed superhydrophobic surface has the characteristics of simple operation and environmental protection.

A Cassie model surface with a rough structure was prepared by introducing silicon dioxide particles onto the surface of 107 liquid silicone rubber by vulcanization at room temperature. In order to study the action law of hydrophobic particles on the surface hydrophobicity of silicone rubber, the chemical functional groups and surface micro-morphology were characterized. Then, the contact angle of the hydrophobic surface was measured by the drop projection method, and the action mechanism of the convex structure on the surface was explained from the angle of a three-phase interface. Finally, we tested the dynamic hydrophobic behavior of the material surface by studying the backward bouncing behavior of droplets at different heights and explained the dynamic hydrophobic mechanism of the material surface from the perspective of energy and force balance. This work provides some practical value for the design and preparation of superhydrophobic surfaces and enriches superhydrophobic theory. Figure 1 shows part of the mechanism diagram and the experimental device diagram of superhydrophobic fluorine rubber surface.

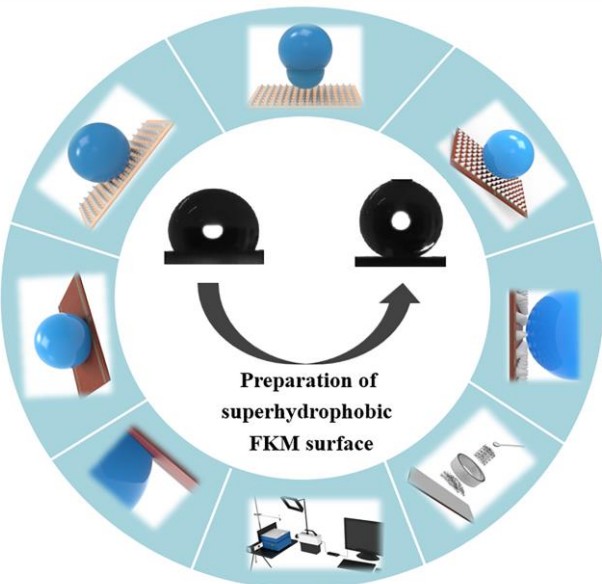

**Figure 1.** Fluororubber surface superhydrophobic.

## 2. Experiment

### 2.1. Materials

Fluorine rubber (FKM): BDF-KK2802CD, Chen guang Bo da Rubber & Plastic Co., LTD. (Chengdu, China), was used as a substrate during the superhydrophobic preparation. Anhydrous ethanol, TianJin Fuyu Chemical Co., LTD, (Tianjin, China), an analytical reagent, was used as a cleaning solution for the superhydrophobic surfaces. An amount of 107 Liquid silicone rubber, a curing agent, Shandong Xingchi Chemical Co., LTD, (Zibo, China), and an analytical reagent, were used. As an adhesive, its curing temperature is room temperature, and there is no humidity requirement. Hydrophobic silicon dioxide (HB-192V), with a purity of more than 98%, Yi chang Hui Rich Silicon Material Co., LTD (Yichang, China), HB-192V, was used as a rough structure for preparing the superhydrophobic surface, in addition to homemade laboratory deionized water.

### 2.2. Preparation of Superhydrophobic Surfaces

Pour around 15 g of 107 liquid silicone rubber into a beaker, along with 3% curing agent, and stir for 3–5 min to make the curing agent and silicone rubber fully integrate. Take out the fluorine rubber sample, lay it flat, pour about 0.6 g silicone rubber into the center of the sample and evenly smear silicone rubber on the surface of the sample with a pin (the original silica gel/fluorine rubber sample (sample A) was obtained after curing for 2 h). Determine the size of the fluorine rubber sample and the amount of silicone rubber poured in order to make the 107 silica gel thin layer left on the fluorine rubber surface have the same thickness. After, cure 107 silicone rubber for about 10 min so that the silicone rubber is in a semi-cured state, and then evenly sprinkle HB-192V evenly on the rubber surface with a screen until the surface is completely covered, and then wait for complete curing. After curing for about two hours, take out the sample and blow dry with nitrogen the remaining HB-192V on the surface. Then, place the sample a beaker filled with anhydrous ethanol and shake with an ultrasonic wave for 5 min and wait for the surface to completely dry. The superhydrophobic surface of fluorine rubber was prepared (sample B), and its preparation flow chart is as Figure 2:

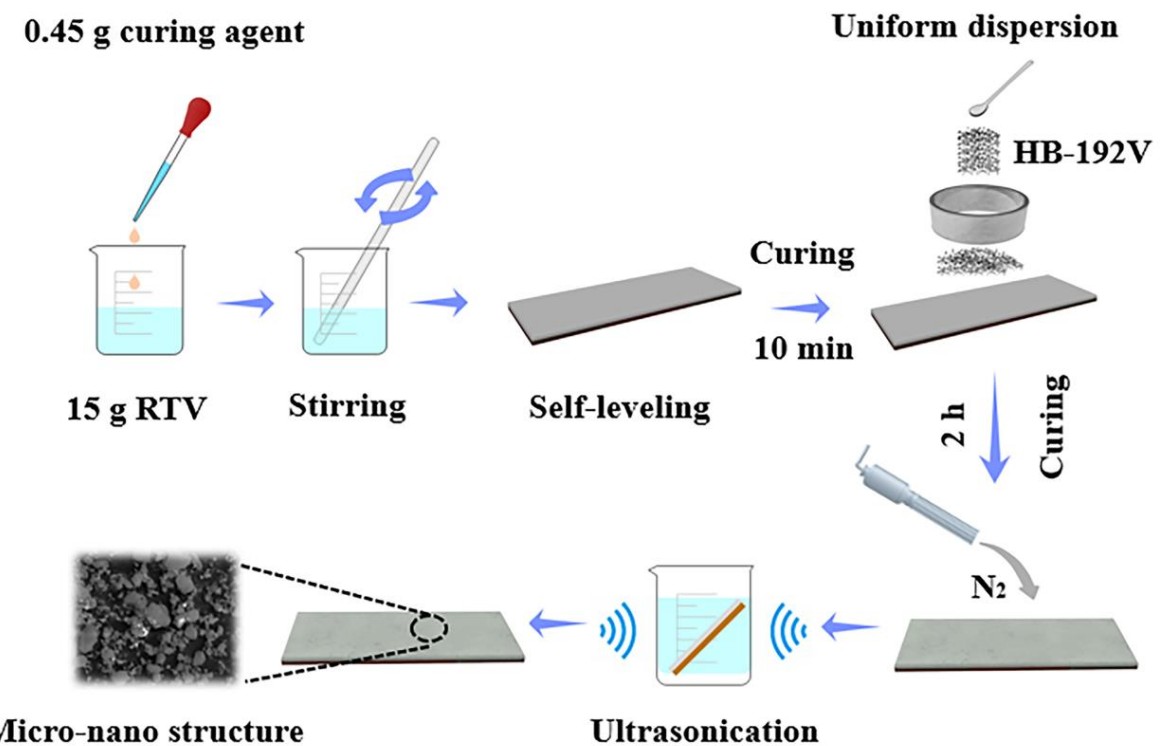

**Figure 2.** Preparation process of fluororubber superhydrophobic surface.

*2.3. Sample Characterization and Performance Test*

Characterization and analysis of particles: Scanning electron microscopy (SEM450, NOVANANO Company (USA)) was used to observe the distribution of particles on the conductive adhesive, and its particle size was measured, and then Fourier-transform infrared spectroscopy was used to analyze the characteristic peaks of the particles and determine the group and main elements of the particles.

Analysis of the samples: The microscopic morphology of the sample surface was observed by scanning electron microscopy (SEM450), and the structures of different sample surfaces were observed under high magnification. EDS analysis was conducted on different surfaces, and the combination of particles and rubber was determined by the changes of the element content on the sample surface. The mechanism of hydrophobicity on different surface contact angles was described.

Measurement of contact angle: A contact angle measuring instrument (HKCA-15, Beijing Hake Test Instrument Factory (Beijing, China)) was used to collect contact angle data from three random measurement points of fluororubber, 107 silica gel, and powder surface, respectively. The droplet size used for contact angle measurement was 4 μL.

Bounce test: The main equipment used in this bounce test was a HX-6E high-speed camera (MEMRECAM Company (Japan)). The experiment is carried out under the lighting of LED film and a television lamp. The size of droplets is controlled by the size of the needle. The droplet volume of the two needles was 5 μL and 10 μL, and the droplet was dropped from four different falling heights of 4 mm, 9 mm, 14 mm, and 19 mm. Combined with the software, the bouncing results of the droplets captured were analyzed. The bouncing diagram is shown in Figure 3.

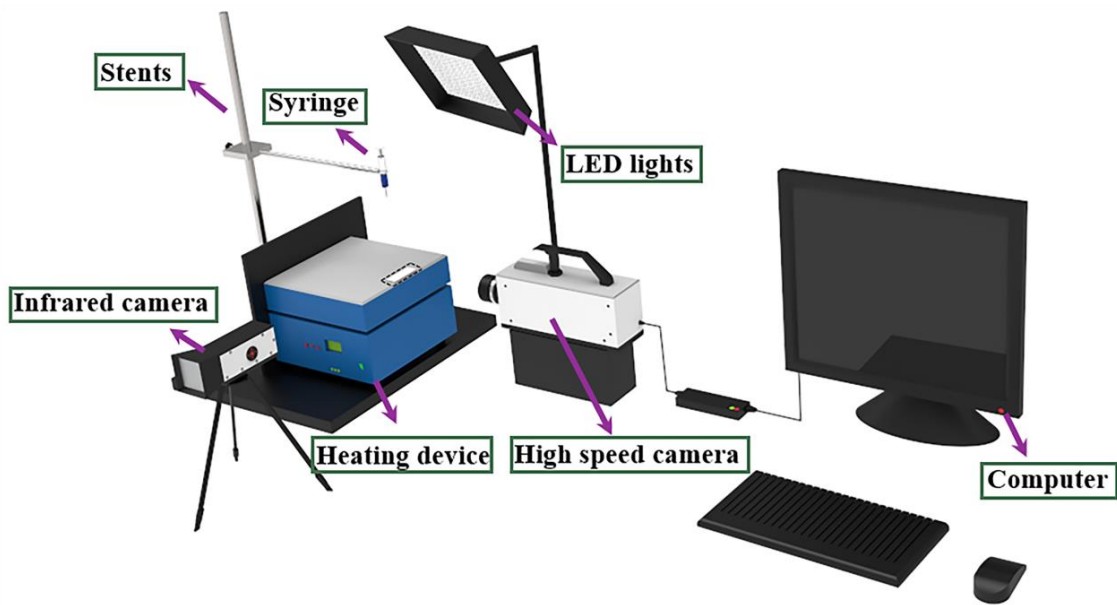

**Figure 3.** Droplet bounce experimental device diagram.

## 3. Results and Discussion

### 3.1. Characterization Analysis of Particles

By observing the electron microscope, we found that the dispersion of HB-192V particles was not good enough and agglomeration could easily occur, so the particles partially aggregated and distributed on the conductive adhesive, as shown in Figure 4a. Relatively independent particles were found in Figure 4a for a local zoomed observation, as shown in Figure 4b. It can be seen that the appearance of HB-192V is not a plane, but a tiny particle with a diameter of about 7.527 μm. When droplets fall on the surface of particles, the contact area between the particles and droplets is greatly reduced due to the non-planar nature of particles, so this microstructure is an important factor to promote the superhydrophobicity of the surface.

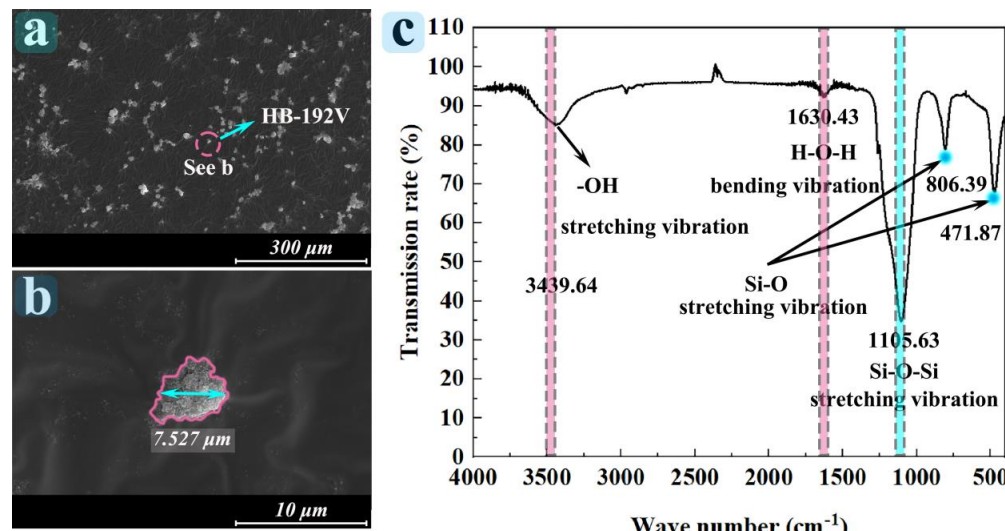

**Figure 4.** (**a**) Shows the macroscopic distribution of HB–192V on conductive adhesive; (**b**) is a local magnification of the particles circled in Figure (**a**); (**c**) Fourier-transform infrared spectra of HB–192V.

By observing the infrared spectrum of the particles in Figure 4c, it can be seen intuitively that there is a strong absorption band at 1105.63 cm$^{-1}$, where the anti-symmetric

stretching vibration of Si-O-Si is. The peaks at 806.39 cm$^{-1}$ and 471.87 cm$^{-1}$ are the symmetric stretching vibration peaks of the Si-O bond. The absorption peaks at 1630.43 cm$^{-1}$ and 3439.63 cm$^{-1}$ correspond to water molecules. Combined with certain literature, we know that the characteristic peaks of silicon dioxide at 1105.63 cm$^{-1}$, 806.39 cm$^{-1}$ and 471.87 cm$^{-1}$ of this product are consistent with the descriptions in the literature [28], and it can be determined that HB-192V is in essence a kind of silicon dioxide.

### 3.2. Characterization of Samples

It is very important for this experiment whether silica powder is distributed on the rubber surface. In order to judge whether HB-192V particles have been coated, an EDS analysis is performed on the sample surface. Because there is a 107 silicone rubber bonding coating between the fluorine rubber surface and 192V, the results can be obtained by analyzing and comparing the 107 silicone rubber surface (sample A) and the fluorine rubber superhydrophobic surface (sample B).

As shown in Figure 5, it can be seen from the energy spectrum that the main elements on the two surfaces are C, O, and Si, but there are some differences in content. The Si content on the surface of sample A and sample B is 48.28% and 47.77%, respectively. The Si content on the surface of the two samples is the most abundant element, and the content is basically the same. There are differences in the contents of C and O elements on the surfaces of the two samples. The content of O element on the surface of sample B is significantly higher than that on the surface of sample A, whereas the content of C element in sample B is significantly lower than that on the surface of sample A. Combined with the infrared spectrum analysis of the particle shown in Figure 4c, the Si powder-containing element is the main element of the O and Si elements. From the content analysis of these elements, we can understand that they are the cause of the change, which is mainly SiO$_2$ particles coated on the surface, causing dusting the surface O element content increased, whereas the coating on the surface of the oxygen content is increased, and the coated particles will cover part A of the sample, resulting in the reduction in C element. The surface of sample B also contains more Si. After coating HB-192V, the Si contained in SiO$_2$ particles balances that on the surface of 107 silicone rubber, so there is basically no significant difference in the content of Si between the two samples. By observing Figure 4c, it can be seen that SiO$_2$ particles are not evenly distributed on the surface of sample B, and there is a phenomenon of regional agglomeration. As seen in Figure 4h, the distribution of element O on the surface of sample B is exactly consistent with the regional agglomeration particles in Figure 4c. Compared with Cheng et al. [36], the superhydrophobic surface of the fluorine-silicon copolymer of the copper–aluminum composite prepared by the spraying method is more complicated than of the simple powder spraying method used in this paper, and the prepared superhydrophobic surface still has the phenomenon of particle agglomeration and large pits. It can be seen from the above phenomenon that HB-192V particles have been successfully coated on the rubber surface, and a microscopic surface with rough structure has been constructed on the rubber.

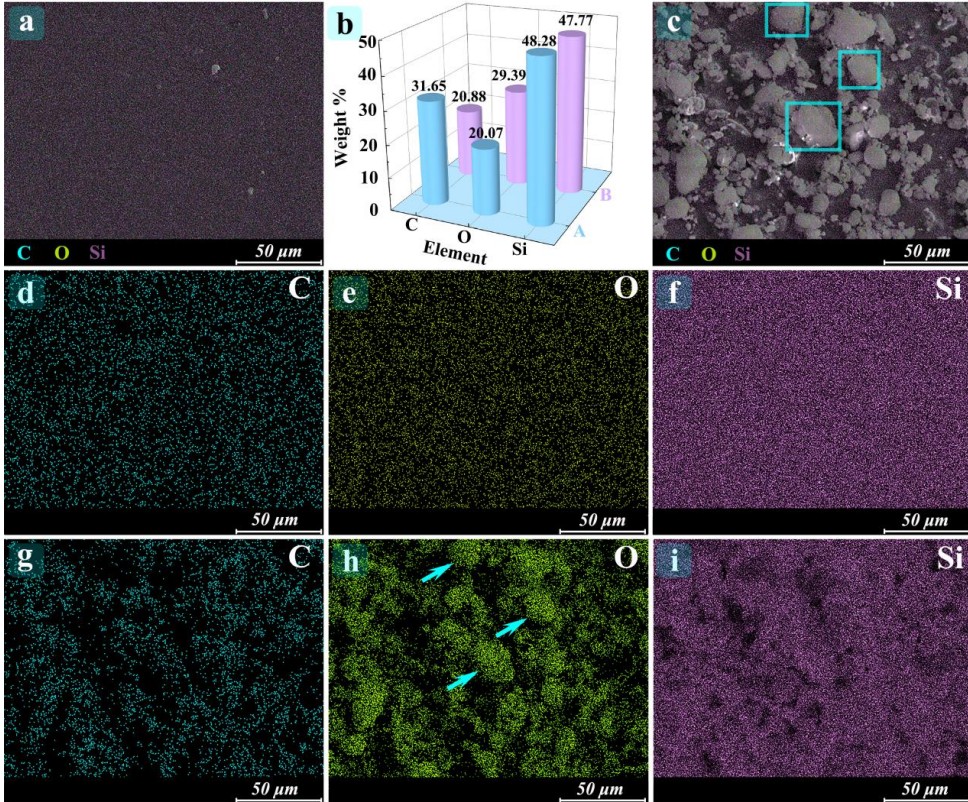

**Figure 5.** (**a**) Shows the surface element distribution of sample A; (**b**) is the bar chart of surface element content of samples A and B; (**c**) is the surface element distribution of sample B; (**d–f**) are respectively energy spectra of elements C, O, Si in the scanning region of energy spectrum of sample A; (**g–i**) are, respectively, energy spectra of elements C, O, Si in the scanning region of energy spectrum of sample B.

The surface of the sample was observed by SEM under the condition of a low magnification. Sample A's surface was relatively smooth, and upon further enlargement of sample A's surface, it was found that the surface was still relatively smooth, and except for a few small areas, the rest of the surface was found to be smooth. Figure 6d,e are scanning electron microscope images of sample B after silicone rubber surface treatment. By observing Figure 6d, it can be seen that $SiO_2$ particles are evenly distributed on the surface of sample B, and a convex structure is generated on the surface of sample B. By observing Figure 6e, it can be seen that $SiO_2$ particles with larger particle size are exposed on the surface of sample B, whereas $SiO_2$ particles with a smaller particle size are embedded in the silicon rubber. Such a special structural distribution greatly increases the roughness of sample B's surface.

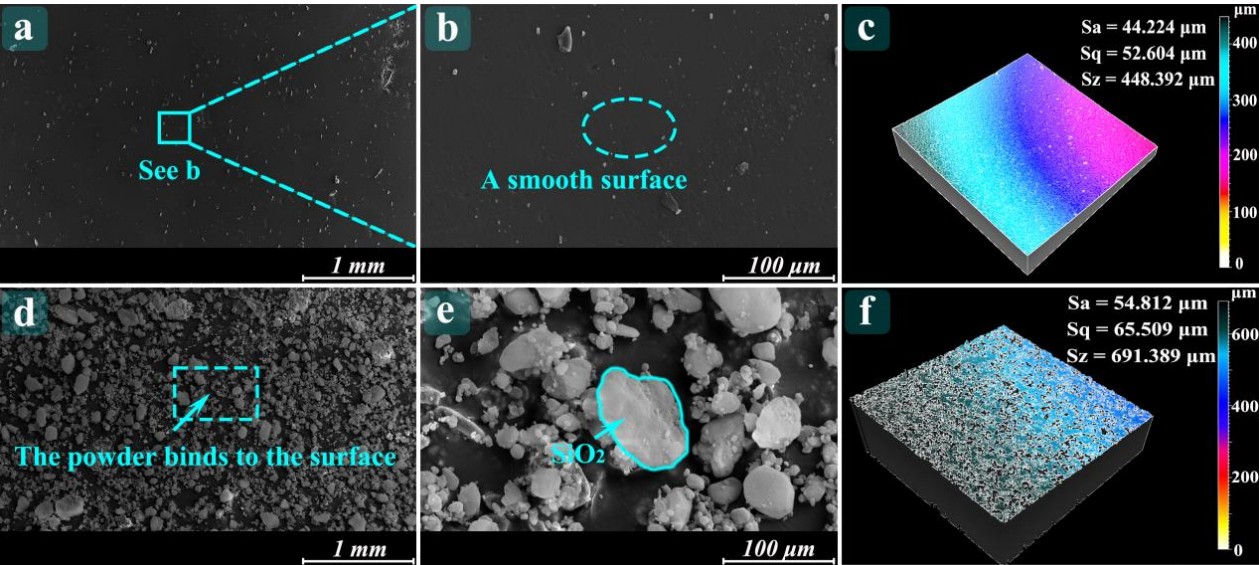

**Figure 6.** (**a**) Shows the SEM image of sample A; (**b**) is an enlarged view of the box area of (**a**); (**c**) is the interferometric three-dimensional topography cloud image of sample A; (**d**) is the scanning electron microscope image of sample B; (**e**) is an enlarged view of the box area in (**d**); (**f**) is the interferometric three-dimensional topography cloud image of sample B.

The surface roughness of sample A and sample B was quantitatively measured by interferometry. Results are shown in Figure 5c,f. The arithmetic average height (Sa) of sample A was 44.224 μm, the root mean square height (Sq) was 52.604 μm, and the maximum height (Sz) was 448.392 μm. The arithmetic mean height (Sa), root mean square height (Sq), and maximum height (Sz) of sample B were 54.812 μm, 65.509 μm and 691 μm, respectively. $SiO_2$ particles are deposited on the silicone rubber surface, which increases the roughness of the silicone rubber surface and forms a micron-level rough structure on the surface. This change reduces the contact area between water droplets and silicone rubber, and the contact state between water droplets and silicone rubber surface changes from Wenzel contact [37] to Cassie contact [38]. Therefore, the surface of sample B has good superhydrophobicity.

### 3.3. Hydrophobicity and Mechanism Analysis

Figure 7a shows the contact angle on the surface of sample A, which is 118.42 ± 1.2°, far from reaching the requirement of superhydrophobic. As shown in Figure 7b, the contact angle of sample B is 154.01 ± 1.1°, which is a superhydrophobic surface. In Cassie's [39] superhydrophobic model, the solid–liquid contact ratio is an important factor affecting superhydrophobic performance [40]. The surface of sample A was not attached or deposited by $SiO_2$ particles, and the surface was relatively smooth compared with sample B, with a smaller contact angle and a larger solid–liquid contact ratio. The droplet on the surface of sample A could hardly roll by overcoming the adhesion force on the silicone rubber surface, and its mechanism is shown in Figure 7c. After the special treatment to the surface of sample B, $SiO_2$ particles have been attached to the surface of the silicone rubber, forming a micron-level rough structure. Due to the attachment of $SiO_2$ particles, numerous small convex structures are formed on the surface of the silicone rubber to support the droplets. Because of the depressions between the convex structures, an air layer is formed. The contact area between the droplet and the silicone rubber surface is greatly reduced, and the droplet itself cannot contact the sag structure formed on the surface of sample B due to the action of surface tension. As a result, the contact angle of the droplet increases, and the droplet easily overcomes the adhesion force on the surface of sample B and rolls.

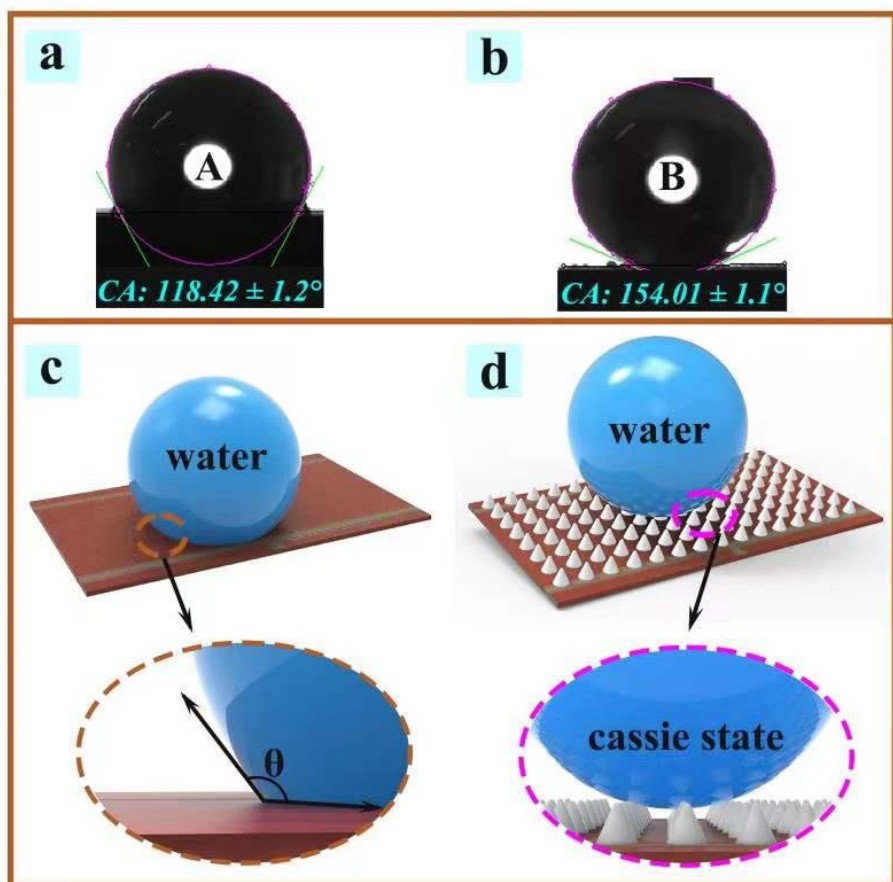

**Figure 7.** (**a**) Contact angle data of sample A's surface; (**b**) is the contact angle data of sample B's surface; (**c**,**d**) are the changes of the sample surface contact model before and after physical deposition of SiO$_2$ particles and their respective regional magnification diagrams.

*3.4. Analysis of Bounce Test*

The bouncing properties of droplets is an important index to evaluate the hydrophobicity of a material's surface. The better the bouncing properties of droplet is, the better the hydrophobicity of material surface is.

In order to study the bouncing behavior of a superhydrophobic surface at room temperature, in this paper, two volumes of droplets were dropped on the surface of the sample at four different heights to carry out the bouncing experiment. Figure 8a,b show the process of 5 μL droplets and 10 μL touching and bouncing off the sample surface for the first time. As can be seen from Figure 8a, when the drop height is 4 mm, the bouncing height of the droplet is only 0.97 mm, as shown in Figure 8e, and at the moment of contact with the sample surface, the droplet is subjected to gravity and the support force provided by the sample surface. The droplet needs to overcome the resistance to work in the process of upward movement. When the falling height is reduced, the residual energy of the droplet after overcoming the resistance to work is very small, so the bouncing height of the droplet is small. According to Figure 8c,d, it can be seen that the maximum spreading diameter of the 10 μL droplet is greater than the 5 μL droplet when the droplet falls at the same falling height. When the 5 μL droplet drops to the surface of the sample, it can be seen that the greater the drop height, the greater the maximum spread diameter and jump height. Interestingly, when the droplet volume is 10 μL, the maximum spread diameter of the droplet increases with the increase in the drop height, but the rebound height of the droplet decreases with the increase in the drop height. This phenomenon is closely related to the stretching phenomenon of the droplet during the bouncing process. Because when the droplet volume is 10 μL, the droplet has a strong stretch before reaching the maximum

bounce height. As shown in Figure 8f, this stretch becomes more intense as the droplet rises. Therefore, at a certain height, the droplet volume is an important factor affecting the droplet bounce height.

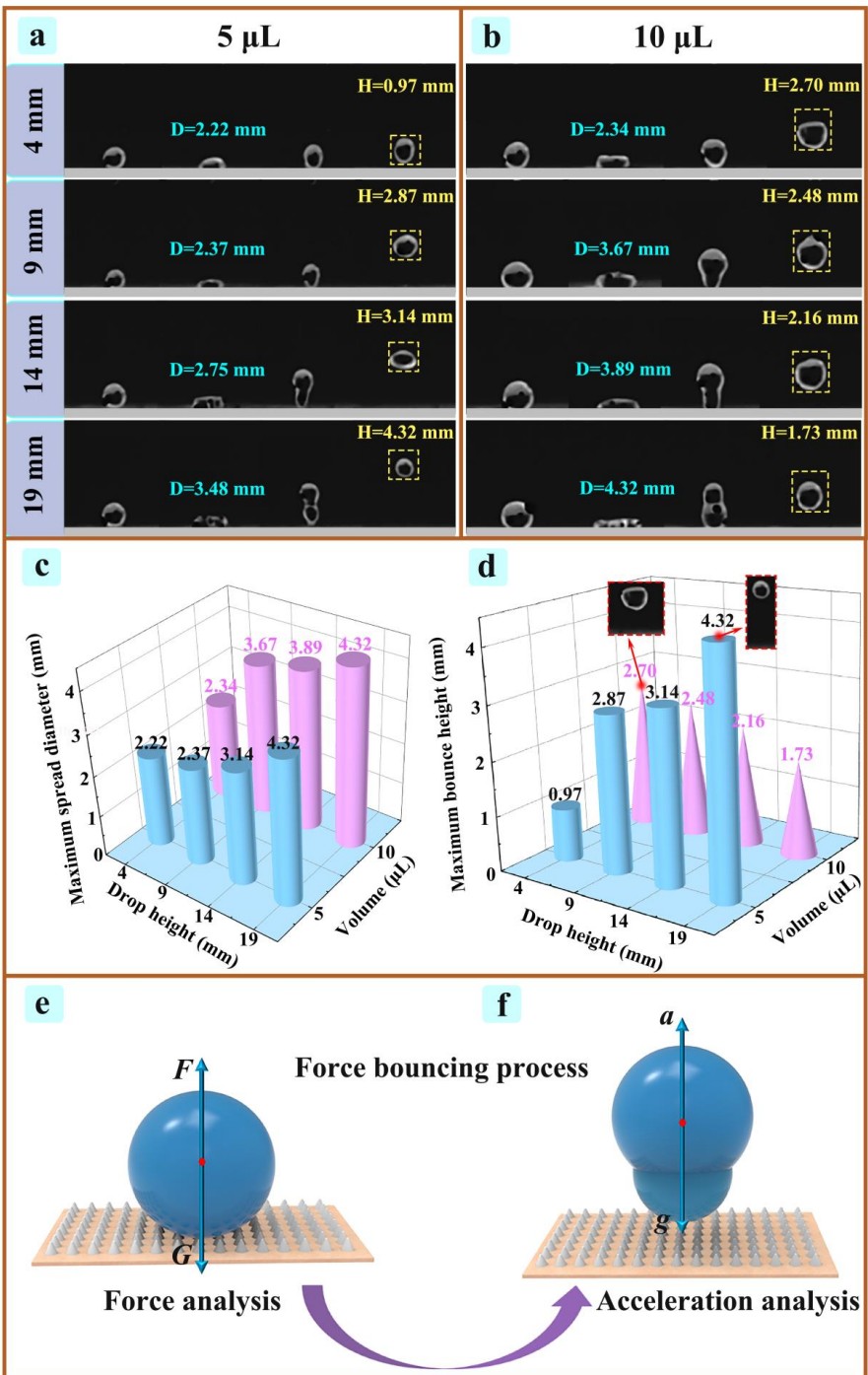

**Figure 8.** (**a**) Shows the bouncing behavior of 5 μL droplet falling from different heights to the surface of sample B at room temperature; (**b**) is the bouncing behavior of 10 μL droplet falling from different heights to the surface of sample B at room temperature; (**c**) is the maximum spreading diameter of the droplet falling from different heights to the surface of sample B under horizontal conditions at room temperature; (**d**) is the maximum bounce height of the droplet falling from different heights to the sample B's superhydrophobic surface under a normal temperature horizontal condition; (**e**) is the force analysis diagram of droplets on the contact sample surface; (**f**) is the acceleration analysis diagram of the droplet in the bouncing process.

## 4. Conclusions

In summary, a micro-rough structure was constructed on the surface of fluorine rubber by a simple physical deposition method, and a superhydrophobic surface with significant self-cleaning performance and droplet bounce performance were prepared. By EDS analysis of the rubber surface, it was found that HB-192V powder can be successfully coated on a fluorine rubber surface on 107 silicone rubber adhesive joints. Through a microstructural analysis of the different surfaces, it is proved that the difference of surface roughness affects the hydrophobicity of surface. The surface roughness of the sample coated with powder is obviously greater than that of fluorine rubber and adhesive 107 silicone rubber. The average contact angle is 154.01°. The current work proves that the surface roughness affects the hydrophobic properties of the material surface. In the bounce test, the surface of the powder has a good bounce performance, and the experimental droplets can regularly bounce and basically do not stick. So, it can be proved that the surface has a good superhydrophobic performance. Because of its excellent mechanical properties and high-temperature resistance, fluorine rubber with superhydrophobic properties will have great application prospects.

**Author Contributions:** X.H.: investigation, writing—original draft preparation; S.R.: data curation, formal analysis. R.T.: supervision. All authors have read and agreed to the published version of the manuscript.

**Funding:** This manuscript has no project support.

**Institutional Review Board Statement:** Not applicable.

**Informed Consent Statement:** Not applicable.

**Data Availability Statement:** Not applicable.

**Conflicts of Interest:** The authors declare no conflict of interest.

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
