# Peer review of "The Preparation of a Superhydrophobic Fluorine Rubber Surface"

_coatings, doi:10.3390/coatings12121878_

Round 1

Reviewer 1 Report

The authors demonstrate super hydrophobic fluorine rubber surface by modifying low surface energy substances as well as rough surface structures. The authors used silica particles (HB-192V) for achieving micro-structured surfaces. After all, the composite film exhibited the contact angle of ~154o, which was evidenced by the measurement of surface wettability and droplet-bounce experimental device. However, the surface modification methods have been reported in lots of previous literatures even including regularly-tuned nano-structured surfaces. In addition, there are insufficient evidences to address the results as follow:

1.    While the top surface consists of the silica particles (HB-192V) with the hydrophilic property, leading to the Wenzel contact than Cassie one, the HB-192V coated surface displays superhydrophobic surface (contact angle of ~154o) and the difference of surface roughness between sample A and B seems to be insignificant. Please explain the reason to have superhydrophobic surface property.

2.    The authors should clearly address the interaction between the silica particle and the substrate surface.

3.    There is no characterization of surface including fluorine rubber.

4.    The authors should provide the chemical structures of used materials for the readers to easily understand.

5.    The authors should explain the novelty of this work compared to the previous literatures reported other superhydrophobic surfaces.

Based on the above consideration, the reviewer can recommend to publish the article in this journal after the major revision.

Author Response

Dear Referees:

Special thanks for the reviewer’s professional comments concerning our manuscript entitled “The preparation of super hydrophobic fluorine rubber surface”. The comments not only promote the quality of the manuscript, but also will play an important role in our later research work. Based on referees’ comments and editor’s decision, we have revised our entire manuscript. The corrections have been included in the revised manuscript and the details are listed as follows.

1.While the top surface consists of the silica particles (HB-192V) with the hydrophilic property, leading to the Wenzel contact than Cassie one, the HB-192V coated surface displays superhydrophobic surface (contact angle of ~154°) and the difference of surface roughness between sample A and B seems to be insignificant. Please explain the reason to have superhydrophobic surface property.

Response: Thanks for the reviewer’s advice. The reason why the surface has superhydrophobic properties can be explained from the following two aspects: First, commercial hydrophobic silica was used to construct a rough structure on the surface of fluororubber. Secondly, Cassie model is formed in the rough surface structure, and many small convex structures support the droplets. Because air pockets are formed between the convex structures, the contact area between the liquid droplets and the rubber surface is greatly reduced, which leads to the contact angle of the rubber surface.

2.The authors should clearly address the interaction between the silica particle and the substrate surface.

Response: Thanks for the reviewer’s advice. As the silica is covered on the surface of liquid silicone rubber through the screen when it is not completely cured, and then the liquid silicone rubber is completely cured, at this time, the silica is semi-immersed in the liquid silicone rubber, so the silica particles and the substrate belong to physical coating. In this paper, the preparation process of superhydrophobic surface has been supplemented.

  1. There is no characterization of surface including fluorine rubber.

Response: Thanks for the reviewer’s advice. In this paper, 107 silicone rubber is used as an intermediate to prepare superhydrophobic surface, and its matrix can also be other solids. Although this paper is used to prepare superhydrophobic surface on fluororubber surface, it is not necessary to characterize the surface of fluororubber, so this paper does not characterize the surface of fluororubber.

  1. The authors should provide the chemical structures of used materials for the readers to easily understand.

Response: Thanks for the reviewer’s advice. As the silica used is a commercial silicon material with no complete chemical structure, its main chain is Si-O-Si, and its side chain functional groups are shown in Figure 4 (c).

  1. The authors should explain the novelty of this work compared to the previous literatures reported other superhydrophobic surfaces.

Response: Thanks for the reviewer’s advice. Compared with the existing construction strategies of superhydrophobic surface, the superhydrophobic surface constructed in this paper has the characteristics of simple operation and environmental protection. Its content has been supplemented in the introduction of this article.

In addition to the above modifications, we have carefully checked and proofread this paper for many times, including grammar, punctuation, tenses, pictures and references. All modifications in the paper are highlighted in yellow. If you have any queries, please don’t hesitate to contact me.

Reviewer 2 Report

The article should be rejected due to the following:

1) Extensive English editing are required. The phrases are constructed with very long sentences (4-6 lines) that hamper scientific comprehension.

2) Low scientific soundness due to lack of results discussion and detail. The majority of results are just presented.

3) More experiments are required in order to support the results and justifications given by the authors.

4) Low number of references that prompt the current state-of-the-art in the given research area. Moreover, the novelty and envisaged applications are not presented.

5) More citations are required to compare the results observed with other literatures. Additionally, a large number of references older than 2017 are referred throughout the manuscript. References list are not uniformly described, information such as year of publication is missing in some cases.

Author Response

Dear Referees:

Special thanks for the reviewer’s professional comments concerning our manuscript entitled “The preparation of super hydrophobic fluorine rubber surface”. The comments not only promote the quality of the manuscript, but also will play an important role in our later research work. Based on referees’ comments and editor’s decision, we have revised our entire manuscript. The corrections have been included in the revised manuscript and the details are listed as follows.

-The authors utilized a large number of references that are older than 2017 (18 references, precisely). Moreover, some of them are incomplete. The year of publication is missing. Please, update the references for newer ones (whenever possible) to enhance its novelty and demonstrate the current status of the state-of-the-art.

Response: Thanks for the reviewer’s advice. References in this paper have been updated and supplemented in this paper, and some of its parameter references are listed as follows:

  1. Zheqin Dong, Maja Vuckovac, Wenjuan Cui, et al. 3D printing of superhydrophobic objects with bulk nanostructure. Advanced materials (Deerfield Beach, Fla.), 2021, 33(45).
  2. Bains Navdeep Singh, Vaishya Rahul O., Kant Suman, et al. Bioinspired hydrophobic surfaces: An overview. Materials Today: Proceedings, 2022, 64(P3).
  3. Jie Liu, Xinwen Zhang, Ruoyun Wang, et al. A mosquito-eye-like superhydrophobic coating with super robustness against abrasion. Materials & Design, 2021, 203.
  4. Yonghua Wang, Zhongbin Zhang, Jinkai Xu, et al. One-step method using laser for large-scale preparation of bionic super-hydrophobic & drag-reducing fish-scale surface. Surface and Coatings Technology, 2021(prepublish).
  5. Zhuoyue Chen, Zhuohao Zhang, Yu Wang, et al. Butterfly inspired functional materials. Materials Science & Engineering R, 2021, 144.
  6. Kaiqiang Zhang, Feng Xu, Yanfeng Gao. Superhydrophobic and oleophobic dual-function coating with durablity and self-healing property based on a waterborne solution. Applied Materials Today, 2021, 22.
  7. Yuting Jin, Longwei Huang, Ke Zheng, et al. Blending electrostatic spinning fabrication of superhydrophilic/underwater su-peroleophobic polysulfonamide/polyvinylpyrrolidone nanofibrous membranes for efficient oil-water emulsion separation. Langmuir : the ACS journal of surfaces and colloids, 2022.
  8. AbuThabit Nedal Y, Uwaezuoke Onyinye J, Abu Elella Mahmoud H. Superhydrophobic nanohybrid sponges for separation of oil/ water mixtures. Chemosphere, 2022, 294.
  9. Eungjun Lee, Do Hyun Kim. Simple fabrication of asphalt-based superhydrophobic surface with controllable wetting transition from Cassie-Baxter to Wenzel wetting state. Colloids and Surfaces A: Physicochemical and Engineering Aspects, 2021, 625.

-The authors should rephrase the abstract section. Longer phrases (>4 lines) must be avoided to enhance reader’s comprehension and logical thinking.

Response: Thanks for the reviewer’s advice. The abstract of this article has been revised in the original text, as follows:

Abstract: Super-hydrophobic materials have a good application prospect in self-cleaning, anti-fouling, anti-corrosion and anti-freezing. However, creating large areas of simple, efficient and environmentally friendly superhydrophobic sur-faces remains a huge challenge. In this paper, using the method of using simple and easy to operate surface dusting, based on 107 silicone rubber adhesive, a super hydrophobic coating with micro/nano structure was constructed on the surface of fluorine rubber. The particle size and groups of HB-192V powder (mainly SiO2) were observed by Fourier transform infrared spectroscopy and scanning electron microscope (SEM). The structures of two kinds of rubber surfaces were observed by SEM, and the superhydrophobic surface materials were qualitatively analyzed by X-ray dif-fraction (XRD). The hydrophobic property of superhydrophobic surface and the bouncing property of droplets were analyzed by contact angle measuring instrument and high-speed camera. The results showed that the preparation of super hydrophobic fluorine rubber on the surface of the water meter contact Angle can reach an average of 154.1°. The superhydrophobic surface prepared by this method has better hydrophobic property and droplet bounce property.

-At the sentence “It has good high temperature resistance, low temperature resistance, corrosion resistance and aging resistance, and good mechanical properties” there is an incongruence. Does it present high or low temperature resistance?

Response: Thanks for the reviewer’s advice. The high and low temperature resistance of fluororubber has not been studied in this paper, but according to the related literature, fluororubber has good high and low temperature resistance. In this paper, the related literature on the high and low temperature resistance of fluororubber has been supplemented.

  1. Jeongbae Park, Beomcheol Lee, Hong Jip Kim. A comparative study on the physical properties of a fluororubber complex. Materials Science and Technology, 2022, 38(18).
  2. Haiyue Zhou, Shikun Li, Zeng Zhang, et al. Preparation of fluororubber/carbon nanotube composites and the effect of carbon nanotubes on aging resistance and solvent resistance of fluororubber. Journal of Macromolecular Science, Part A, 2022, 59(10).

-In the Introduction section it is not clear the current state-of-art in the research area. The authors mentioned and justified the type of material utilized in the development of superhydrophobic platforms however, it is poorly compared with only another research group. Therefore, it might induce the reader that the manuscript is the first one regarding the subject. If it is the case, it should be highlighted throughout the manuscript. More references regarding different groups are required to better compose the literature background. As a result, the authors are encouraged to fully differentiate its work and point out its novelty and envisaged applications.

Response: Thanks for the reviewer’s advice. In the introduction, new references have been supplemented, and the previous work has been analyzed and summarized.

-The figure 1 is not clear and does not transmit a logical idea of the superhydrophobic surface preparation.

Response: Thanks for the reviewer’s advice. The function of Figure 1 is to make readers better understand the article, which is equivalent to the function of a graphic summary. It only shows part of the mechanism diagram and experimental device diagram of superhydrophobic fluororubber surface, not the preparation flow chart of superhydrophobic surface. The flow chart of superhydrophobic preparation of fluororubber surface is shown in Figure 2.

-The description of each utilized material is not complete. It lacks some clear information such as purity, molar weight and role (for example). What is the compound HB-192V? The curing was performed under what conditions of temperature and humidity? It must be mentioned in the manuscript.

Response: Thanks for the reviewer’s advice. In the material part of this paper, the materials used have been supplemented, as follows:

Fluorine rubber (FKM): BDF-KK2802CD, Chen guang Bo da Rubber & Plastic Co., LTD. (Chengdu, China), which is used as a substrate during superhydrophobic prepa-ration. Anhydrous ethanol, Fu yu Chemical Co., LTD, (China), analytical reagent, used as cleaning solution for superhydrophobic surfaces. 107 Liquid silicone rubber, curing agent, Shan dong Xing chi Chemical Co., LTD, (China), analytical reagent. As an adhesive, its curing temperature is normal temperature, and there is no humidity require-ment. Hydrophobic silicon dioxide (HB-192V) with a purity of more than 98%, Yi chang Hui Rich Silicon Material Co., LTD, HB-192V as a rough structure for preparing superhydrophobic surface; Laboratory homemade deionized water.

-The authors referred to the developed material as conductive adhesive. What is conductive state for? Does it mean electro conductive?

Response: Thanks for the reviewer’s advice. The conductive adhesive mentioned in this paper is used to fix the sample on the sample table more stably and scan the sample more clearly during the scanning electron microscope. Not the materials used are conductive.

-What was the average dimensions of the final developed materials? This information needs to be stated throughout the manuscript. Please, also provide an image (with scale) of the developed material.

Response: Thanks for the reviewer’s advice. The average size of the materials used in this paper is 7.527 μm, and the SEM images of the particles have been shown in Figure 4.

-The number of figures throughout the text are missing. Please correct it.

Response: Thanks for the reviewer’s advice. The figures in the picture have all been supplemented in the text.

-What was the justification of the statement “the surface shows tiny honeycomb holes. Such microstructure is an important factor promoting the surface to achieve superhydrophobicity”? The result must be clearly discussed in detail.

Response: Thanks for the reviewer’s advice. "There are tiny honeycomb holes on the surface" is a nonstandard expression, which has been revised in the original text, and supplemented the possible influence of the microstructure of particle surface on the construction of superhydrophobic surface. Its specific modifications are as follows:

As can be seen from Fig. 4 (b), HB-192V is a tiny particle with a diameter of about 7.527 μm instead of a plane. When the droplet drops on the surface of the particle, the contact area between the particle and the droplet is greatly reduced due to the non-plane nature of the particle surface, so this microstructure may be an important reason for promoting the superhydrophobicity of the surface.

-References are missing to complement the discussion of FTIR results. Why have the authors not performed the same analysis onto fluorinated rubber? Is there a logical justification?

Response: Thanks for the reviewer’s advice. In this paper, the results of FTIR discussion of particles have been supplemented with relevant references, see Ref. 28 for details. In the preparation process of superhydrophobic surface in this paper, fluororubber only serves as a matrix and does not play a major role, so the same analysis of fluororubber is not carried out in this paper.

  1. Zhang L, Wang S X. Application of rubber products in oil drilling and production engineering. China Rubber, 2011, 27(03): 18-20.

-The description of the results in the sections 3.2-3.4 lack technical information and discussion. The results are just presented but not discussed in detail. Moreover, the authors did not compare the results with other literatures.

Response: Thanks for the reviewer’s advice. Section 3.2-3.4 of this paper has supplemented the description of relevant results, and made a comparative analysis with other related works in the literature.

-Have the authors considered the analysis of contact angles with different media to properly infer superhydrophobicity? Moreover, what about the surface energy values? And the influence of temperature and PH?

Response: Thanks for the reviewer’s advice. As a measuring medium, distilled water or deionized water is mostly used to infer the superhydrophobic contact angle. Therefore, only deionized water is used as the medium in this paper, and the superhydrophobic property of rubber surface can also be expressed. The surface energy of sample A is 12.73 mN/m and that of sample B is 0.61 mN/m measured by contact angle meter. As the influence of temperature and PH value on superhydrophobic surface belongs to durability, further analysis and research will be carried out when experimental conditions are available in the future.

-The authors state that the developed materials are superhydrophobic with a rough micro structured surface. However, more techniques are required to further complement the results and validate it (e.g. AFM, perthometer, CA with different media, surface energy calculations etc.)

Response: Thanks for the reviewer’s advice. Because the application environment of the prepared superhydrophobic surface is natural environment, the medium used is only deionized water. In the future, when there are experimental conditions, the results of analysis and verification will be further supplemented.

In addition to the above modifications, we have carefully checked and proofread this paper for many times, including grammar, punctuation, tenses, pictures and references. All modifications in the paper are highlighted in yellow. If you have any queries, please don’t hesitate to contact me.

Reviewer 3 Report

      The authors present an article that from the beginning of its reading I realize what the result will be without reading further. In this regard I ask, what is the relevance of your manuscript?

The authors need a big improvement in the writing of the English.

Could you indicate what are figures (a), (c), (d) and (f)?

The discussion of Figure 4 is not indicated in the text.

It is not necessary to give an extensive explanation about the free water obtained by infrared, just to indicate that the spectroscopic signal that appears in ... is sufficient.

Measuring the particle size by sem is semi-quantitative, could you do the particle size by BET and/or X-ray microtomography?

Check all the references because they do not comply with what is established by Coatings. In this context, avoid citing by block of references. What do you mean when you indicate: He[27-30]et al. prepared hydrophobic surfaces? Reference [36] does not correspond to what is established since the paper does not indicate contact angles. 

Author Response

Dear Referees:

Special thanks for the reviewer’s professional comments concerning our manuscript entitled “The preparation of super hydrophobic fluorine rubber surface”. The comments not only promote the quality of the manuscript, but also will play an important role in our later research work. Based on referees’ comments and editor’s decision, we have revised our entire manuscript. The corrections have been included in the revised manuscript and the details are listed as follows.

The authors present an article that from the beginning of its reading I realize what the result will be without reading further. In this regard I ask, what is the relevance of your manuscript?

Response: Thanks for the reviewer’s advice. The main work of this paper is to prepare superhydrophobic coating on the surface of fluororubber, and the abstract and introduction of this paper have been supplemented and modified.

The authors need a big improvement in the writing of the English.

Response: Thanks for the reviewer’s advice. In this paper, we have re-checked the grammar and expression, and revised the relevant terms.

Could you indicate what are figures (a), (c), (d) and (f)?

Response: Thanks for the reviewer’s advice. Figures (a), (c), (d) and (f) are written incorrectly, and have been supplemented and modified in this paper.

The discussion of Figure 4 is not indicated in the text.

Response: Thanks for the reviewer’s advice. The discussion on the characterization of the particles used in Figure 4 has been supplemented in the text.

It is not necessary to give an extensive explanation about the free water obtained by infrared, just to indicate that the spectroscopic signal that appears in ... is sufficient.

Response: Thanks for the reviewer’s advice. In this paper, the explanation of infrared spectrum of free water has been modified.

Measuring the particle size by sem is semi-quantitative, could you do the particle size by BET and/or X-ray microtomography?

Response: Thanks for the reviewer’s advice. After fitting the XRD curve with MDI Jade6 software, the half-peak width parameter is calculated, and the particle size of the test particle is about 10 nm after being brought into Scheler formula. It shows that the particle size measured by SEM is the result of particle agglomeration. Of course, the accuracy of particle size measured by software fitting needs to be further confirmed.

Check all the references because they do not comply with what is established by Coatings. In this context, avoid citing by block of references. What do you mean when you indicate: He[27-30]et al. prepared hydrophobic surfaces? Reference [36] does not correspond to what is established since the paper does not indicate contact angles.

Response: Thanks for the reviewer’s advice. In this paper, the inconsistent references have been revised, and the cited inappropriate references have been deleted.

In addition to the above modifications, we have carefully checked and proofread this paper for many times, including grammar, punctuation, tenses, pictures and references. All modifications in the paper are highlighted in yellow. If you have any queries, please don’t hesitate to contact me.

Round 2

Reviewer 1 Report

The authors provided the response point-by-point. The reviewer recommend the revised manuscript in the journal of Coatings.

Author Response

Comments and Suggestions for Authors

The authors provided the response point-by-point. The reviewer recommend the revised manuscript in the journal of Coatings.

Response:Thank the reviewers for their valuable comments on the paper

Reviewer 2 Report

Dear authors,

In the first round I have rejected the given manuscript. In my opinion, the work lacks scientific novelty and does not significantly contribute to the current state-of-the-art.

Author Response

Comments and Suggestions for Authors

Dear authors,

In the first round I have rejected the given manuscript. In my opinion, the work lacks scientific novelty and does not significantly contribute to the current state-of-the-art.

Response:Thank the reviewers for their valuable comments on the paper. I have reorganized the innovation of the paper, and I hope you can recognize it.

Reviewer 3 Report

Dear authors,

I welcome your manuscript with the proposed changes so that the audience will be interested in reading it.

You also included some references and others have been deleted, which I have reviewed carefully.

For my part, I suggest that your manuscript be published.

Author Response

Comments and Suggestions for Authors

Dear authors,

I welcome your manuscript with the proposed changes so that the audience will be interested in reading it.

You also included some references and others have been deleted, which I have reviewed carefully.

For my part, I suggest that your manuscript be published.

The authors provided the response point-by-point. The reviewer recommend the revised manuscript in the journal of Coatings.

Response:Thank the reviewers for their valuable comments on the paper